# Comparison between Colorimetric In Situ Hybridization, Histopathology, and Immunohistochemistry for the Diagnosis of New World Cutaneous Leishmaniasis in Human Skin Samples

**DOI:** 10.3390/tropicalmed7110344

**Published:** 2022-11-01

**Authors:** Luiz Cláudio Ferreira, Leonardo Pereira Quintella, Armando de Oliveira Schubach, Luciana de Freitas Campos Miranda, Maria de Fátima Madeira, Maria Inês Fernandes Pimentel, Érica de Camargo Ferreira e Vasconcellos, Marcelo Rosandiski Lyra, Raquel de Vasconcellos Carvalhaes de Oliveira, Rodrigo Caldas Menezes

**Affiliations:** 1Anatomic Pathology Service, Evandro Chagas National Institute of Infectious Diseases, Oswaldo Cruz Foundation, Av. Brasil, 4365, Rio de Janeiro 21040-360, Brazil; 2Laboratory of Clinical Research and Surveillance of Leishmaniasis, Evandro Chagas National Institute of Infectious Diseases, Oswaldo Cruz Foundation, Av. Brasil, 4365, Rio de Janeiro 21040-360, Brazil; 3Fundação Técnico Educacional Souza Marques, Av. Ernani Cardoso, 335, Rio de Janeiro 21310-310, Brazil; 4Laboratory of Clinical Epidemiology, Evandro Chagas National Institute of Infectious Diseases, Oswaldo Cruz Foundation, Av. Brasil, 4365, Rio de Janeiro 21040-360, Brazil; 5Laboratory of Clinical Research on Dermatozoonoses in Domestic Animals, Evandro Chagas National Institute of Infectious Diseases, Oswaldo Cruz Foundation, Av. Brasil, 4365, Rio de Janeiro 21040-360, Brazil

**Keywords:** tegumentary leishmaniasis, diagnosis, *Leishmania braziliensis*, in situ hybridization, immunoperoxidase

## Abstract

New world cutaneous leishmaniasis (NWCL) is an anthropozoonosis caused by different species of the protozoan *Leishmania.* Colorimetric in situ hybridization (CISH) was shown to satisfactorily detect amastigote forms of *Leishmania* spp. in animal tissues, yet it was not tested for the diagnosis of human NWCL. The aim of this study was to compare CISH, histopathology (HP), and immunohistochemistry (IHC) techniques to diagnose NWCL in human cutaneous lesions. The sample comprised fifty formalin-fixed, paraffin-embedded skin biopsy specimens from patients with NWCL caused by *L. (V.) braziliensis*. These specimens were analyzed by CISH, using a generic probe for *Leishmania*, IHC, and HP to assess the sensitivity of these methods by using a parasitological culture as a standard reference. Additional specimens from three patients diagnosed with cutaneous mycoses were also included to evaluate cross-reactions between CISH and IHC. The sensitivities of IHC, CISH, and HP for detecting amastigotes was 66%, 54%, and 50%, respectively. IHC, unlike CISH, cross-reacted with different species of fungi. Together, these results demonstrate that CISH may be a complementary assay for the detection of amastigote in the laboratorial diagnosis routine of human NWCL caused by *L.* (*V*.) *braziliensis*.

## 1. Introduction

American tegumentary leishmaniasis (ATL) is an anthropozoonosis caused by the new world species of the protozoan *Leishmania,* which causes cutaneous lesions (new world cutaneous leishmaniasis, NWCL) and/or upper aerodigestive tract mucosa (new world mucosal leishmaniasis) in humans and other domestic and wild mammals [1,2,3,4]. It is endemic in 18 countries across the American continent, where an average of 55,000 cases of this disease are recorded per year, notably in Brazil, Colombia, Nicaragua, and Peru [4].

In Brazil, ATL is present in all states, and seven species of the *Leishmania* causing this disease have already been identified, with *Leishmania* (*Viannia*) *braziliensis* being the most common [1,2]. The transmission of ATL-causing species of *Leishmania* to humans and animals occurs through the bite of infected female sand flies of the genus *Lutzomyia* [2].

Laboratory tests that detect the parasite in the lesion are essential to confirm the diagnosis, since epidemiological history, clinical aspects of the lesions, and positivity in immunological methods used, such as the Montenegro intradermal reaction test (IDRM) and serological tests, are only assumptive [2,5]. Certainty in the diagnosis is primary as the drugs used in the treatment of NWCL present high toxicity [6].

Parasitological culture is the reference standard test for the diagnosis of NWCL. Nevertheless, this technique has sensitivity limitations for fresh specimens, as well as the other laboratory methods used, especially when there are small amounts of the parasite in the lesions, which is common in NWCL caused by species of the subgenus *Viannia* [7]. Polymerase chain reaction (PCR) is the most accurate technique for the diagnosis of NWCL [5], despite inaccuracies in sensitivity, notably in cases of strong cellular immune response [8] or when performed in 10% formalin-fixed paraffin-embedded tissues (FFPE) [9]. Histological techniques including histopathology (HP) and immunohistochemistry (IHC) and direct examination are greatly useful in the diagnosis of NWCL, mainly due to their high specificity and good sensitivity as regards IHC [5,10]. However, these techniques also present inaccuracies in sensitivity and, despite having high specificity, false-positive results can occur, especially for histomorphologically similar fungi such as *Histoplasma* spp. and *Sporothrix* spp. [2,10]. Thus, the development of new, more sensitive, and specific diagnosis methods is paramount to fill the existing gaps in the current tests.

To improve the accuracy in histological assays, in situ hybridization applied to FFPE tissues has shown to be promising, for it detects specific nucleic acid segments (DNA or RNA) of *Leishmania* [11,12,13]. In colorimetric in situ hybridization (CISH), an antigen-labeled nucleic acid probe is used, which specifically binds to a complementary nucleic acid sequence of the infectious agent. This binding is detected via an antigen–antibody interaction, which is visualized under light microscopy using chromogenic detection with a colored enzyme substrate [14]. In turn, in fluorescent in situ hybridization (FISH), the probe is fluorochrome-labeled and seen only under a fluorescence microscope [15].

CISH, using a specific probe for both the *L. donovani* complex and the genus *Leishmania*, showed good sensitivity and specificity, superior to those of HP and IHC when tested in skin samples from dogs infected with *L. infantum* [13]. Still, CISH for the detection of the amastigote forms of *Leishmania* spp. in FFPE cutaneous lesion samples for the diagnosis of human NWCL has not yet been evaluated. Therefore, this study aims to evaluate the performance of CISH compared with HP and IHC for the diagnosis of human NWCL caused by *L.* (*V.*) *braziliensis*.

## 2. Materials and Methods

### 2.1. Sample

A convenience sample of 50 specimens of cutaneous lesions was used, collected by biopsy from patients diagnosed with NWCL with an isolation of *Leishmania* in parasitological culture and its subsequent characterization as *L.* (*V.*) *braziliensis*, through multilocus enzyme electrophoresis. Samples collected between 2000 and 2005 were retrospectively selected from a database with information on patients seen at the Evandro Chagas National Institute of Infectious Diseases, Oswaldo Cruz Foundation. The skin samples were fixed in 10% buffered formalin and paraffin-embedded between 2000 and 2005, and were processed by CISH, HP, and IHC in 2017. One of the skin samples was from 2000, another from 2001, eight were from 2002, ten from 2003, 18 from 2004, and 12 from 2005.

The FFPE samples of cutaneous lesions collected by biopsy from three patients diagnosed with cutaneous infections caused by *Sporothrix* sp., *Candida* sp. and *Histoplasma* sp., respectively, were examined and confirmed by histopathology and fungal culture in order to assess whether CISH and IHC showed cross-reaction or cross-hybridization with fungi that cause cutaneous mycoses.

### 2.2. Parasite Culture and Isoenzyme Characterization

Tissue fragments collected by biopsy were conditioned, immediately immersed in a sterile saline solution containing antimicrobials, and then cultured at 26–28 °C in biphasic culture medium Novy-MacNeal-Nicolle plus Schneider’s Drosophila medium (Sigma-Aldrich, St. Louis, MO, USA), supplemented with 10% fetal bovine serum, penicillin, and streptomycin, according to the protocol described at https://dx.doi.org/10.17504/protocols.io.22tggen (accessed on 17 September 2022). The isolated promastigote forms of *Leishmania* were identified as *L.* (*V.*) *braziliensis* by multilocus enzyme electrophoresis [16].

### 2.3. Histopathology

Two serial sections (5 µm) of the paraffin blocks containing the tissues were deparaffinized with xylene, rehydrated in decreasing concentrations of ethanol, and then stained with hematoxylin-eosin (HE) [17].

### 2.4. Immunohistochemistry

Two serial sections (5 µm) of the paraffin blocks containing the tissues were placed onto silanized slides and processed according to Oliveira et al. [18]. Shortly thereafter, the sections were deparaffinized with xylene, rehydrated in decreasing concentrations of ethanol, and submitted to an endogenous peroxidase blockade with a 30% hydrogen peroxide and methanol solution for 40 min at room temperature and protected from light. Antigen retrieval was performed in sodium citrate buffer (pH = 6.0) at 65 °C in a water bath for 30 min. To block nonspecific binding, a protein-blocking serum (Ultra V Block, Lab Vision^TM^, Thermo Scientific, Fremont, CA, USA) was used for 10 min at room temperature. Subsequently, the sections were incubated overnight with polyclonal rabbit anti-*Leishmania* sp. serum obtained in-house according to the protocol described by Quintella et al. [10] and diluted at 1:500. The reaction to detect amastigotes was developed using The HiDef Detection^TM^ HRP Polymer System Kit (Cell Marque, Rocklin, CA, USA), according to the protocol recommended by the manufacturer. The enzyme-substrate/chromogen reaction was developed using diaminobenzidine (DAB) and hydrogen peroxide (Sigma-Aldrich, St. Louis, MO, USA). The sections were counterstained with Harris hematoxylin for 2 min and mounted onto Entellan^TM^ mounting medium (Merck KGaA, Darmstadt, Germany). Histological sections of tissues intensely parasitized with amastigote forms of *Leishmania* sp. And were incubated with non-immune homologous serum as the negative control and with polyclonal rabbit anti-*Leishmania* sp. serum as the positive control.

### 2.5. Colorimetric In Situ Hybridization

Two serial sections (5-µm) of the paraffin blocks containing the tissues were placed onto silanized slides. These sections were then processed according to the protocol by Boechat et al. [19] using an oligonucleotide probe specific for the genus *Leishmania*, digoxigenin-labeled at the 5′-end that targets the 5.8S ribosomal RNA gene of the parasite [11]. Briefly, sections were deparaffinized with xylene and rehydrated in decreasing concentrations of ethanol. The ZytoFastPlus CISH implementation kit AP-NBT/BCIP^®^ (Zytovision GmbH, Bremerhaven, Germany) was used. The proteolytic treatment of the sections was performed using pepsin for 5 min at 37 °C. Next, the sections underwent cell conditioning using sodium citrate buffer (pH = 6.0) at 98 °C in a water bath for 30 min. Later, the sections were dehydrated in increasing concentrations of ethanol and dried in an oven at 37 °C for 10 min. The sections were then incubated using the probe diluted 1:1000 in buffer solution (Hybridization Solution H7782^®^, Sigma-Aldrich, St. Louis, MO, USA) in the ThermoBrite^®^ hybridizer (StatSpin, Westwood, MA, USA). In the hybridizer, the sections were submitted to denaturation at 75 °C for 5 min, followed by overnight hybridization at 37 °C. The following day, the sections underwent three stringency baths in a Tris-buffered-saline solution at a concentration of 1X: the first at room temperature for 5 min; the second at 55 °C for 5 min; and the third at room temperature for 5 min. Following the stringency baths, the sections were incubated with anti-digoxigenin monoclonal antibodies produced in rabbits for 30 min at 37 °C. Then, the sections were incubated with polymer-conjugated anti-rabbit antibody and alkaline phosphatase for 30 min at 37 °C. Subsequently, the sections were submitted to signal visualization using 5-bromo-4-chloro-3 indolyl phosphate (BCIP) and 4-nitro blue tetrazolium chloride (NBT) for 30 min at 37 °C in a dark chamber. Afterwards, the sections were counterstained with fast red nuclear dye for 5 min and mounted onto Entellan^TM^ mounting medium (Merck KGaA, Darmstadt, Germany). The histological sections of the tissues intensely parasitized with amastigote forms of *L. infantum* were incubated using the probe as positive control and hybridization solution as the negative control.

### 2.6. Slide Reading

Microscopic interpretation of CISH, histopathology, and IHC techniques were performed by three observers. Observer 1 was an experienced physician pathologist in the histopathological and immunohistochemical diagnosis of infectious diseases, especially human NWCL. Observer 2 was a veterinary pathologist also experienced in the histopathological, immunohistochemical, and CISH diagnosis in leishmaniasis. Observer 3 was a biologist with laboratory experience in the histopathological and immunohistochemical diagnosis of leishmaniasis, and was responsible for carrying out the IHC and CISH techniques. All slides were examined under an optical microscope using 40 and 100× objective lenses to search for amastigote forms of *Leishmania*, and the final results of the readings were obtained by observer 1, the most experienced.

In HP, the tissue samples stained by HE for the detection of amastigotes were considered positive in the presence of at least one structure with typical morphotintorial aspects (size 1 to 4 µm diameter, staining affinity of nucleus, cytoplasm, and kinetoplast, and round-shaped or pyriform), and were also localized inside the parasitophorous vacuoles of macrophages. The samples processed by IHC and CISH were considered positive when at least one structure morphologically compatible with a brown-stained amastigote form in IHC and a dark-blue-stained one in CISH was observed. In addition, shape, size, texture, and location inside the parasitophorous vacuoles of macrophages were considered. For fungal-containing tissue samples to be considered positive for cross-reactivity or cross-hybridization, both characteristics must be present: morphologically compatible with yeasts or hyphae and brown or dark blue stained by the chromogen used in IHC and CISH, respectively.

### 2.7. Statistical Analysis

Data were analyzed using R Project for Statistical Computing for Windows, software version 4.1.1 [20]. The results by observer 1 (the most experienced) were used to calculate the sensitivity of each technique. The sensitivity values and the 95% confidence intervals (95% CI) for each technique obtained by observer 1 were calculated using positive parasitological culture for *L.* (*V.*) *braziliensis* as the standard reference technique. The Cohen kappa index, the rating of the degree of agreement according to Landis & Koch [21], and the total percentage agreement were provided for the assessment of inter-observer agreement in each technique and amid techniques based on the readings by observer 1.

### 2.8. Ethics Statement

The current study was approved by the Human Subject Research Ethics Committee (CEP) from INI, Fiocruz (under protocol number 51629615.0.0000.5262).

## 3. Results

The sensitivity values for the detection of amastigotes found by IHC, CISH, and HP techniques and the number of positive and negative samples for each of these techniques assessed are described in Table 1.

Table 2 shows the frequency of positivity for the IHC, CISH, and HP techniques in the samples concerning the storage time of the paraffin blocks.

Figure 1 shows positive staining for amastigotes in the cytoplasm of macrophages in the cutaneous lesions analyzed by CISH (Figure 1A) and IHC (Figure 1B). Figure 1C shows the amastigote forms identified by HP.

Table 3 presents the comparison of the three techniques studied in pairs according to the distributions of positive and negative cases for the amastigote forms.

The agreement between the techniques is shown in Table 4 and Table 5.

The inter-observer agreement for IHC, CISH, and HP is shown in Table 6.

CISH showed no cross-hybridization with structures morphologically compatible with yeasts or hyphae of *Sporothrix* sp., *Candida* sp., and *Histoplasma* sp., whereas IHC showed cross-reactions with structures morphologically compatible with yeasts or hyphae of the three fungal species herein evaluated (Figure 2).

## 4. Discussion

A study was performed to compare the sensitivities of three different techniques in the diagnosis of NWCL in patients with diagnostic confirmation by the isolation of *Leishmania* sp. in culture medium, characterized as *L.* (*V.*) *braziliensis*. In this study, IHC showed greater sensitivity when compared with CISH and HP. Although different Cohen kappa index values were obtained among IHC, CISH, and HP, the agreement among these techniques was considered moderate, being a total agreement of around 70%, and the inter-observer agreement ranged from moderate to almost perfect.

IHC showed sensitivity within the range of values previously described by other authors for the diagnosis of human NWCL, ranging from 51 to 76% [5,22,23,24,25,26,27]. The IHC protocols and sample selection criteria in the aforementioned studies were different from those in the present study, which may have influenced the sensitivity variation observed herein.

Nonetheless, according to Quintella et al. [10] and Alves et al. [28], IHC reported higher sensitivity values for the diagnosis of human NWCL. Quintella et al. [10] described a sensitivity of 80% in IHC, using a group of samples similar to that of this study, with a positive culture medium for *Leishmania* of the subgenus *Vianna* as a standard reference technique. However, a different protocol from the present study was used, with heat treatment for antigen retrieval at 100 °C and a streptavidin-biotin-peroxidase detection kit. In this study, a biotin-free polymer-based detection kit and heat treatment for antigen retrieval at 65 °C were employed. It is noteworthy that Quintella et al. [10] used a lower sampling rate (N = 30), which might have influenced the sensitivity results. Moreover, the difference in sensitivities between the studies may also be attributed to a possible difference in the intensity of the parasite load in human NWCL lesions, as well as the different protocol used. Alves et al. [28] found a sensitivity of 92% in IHC in the diagnosis of human NWCL. Due to the fact that these authors used anti-*Leishmania* canine hyperimmune serum as the primary antibody and tested IHC in skin samples that were positive in microscopy and PCR and from patients that were positive for the Montenegro intradermal reaction the high sensitivity value found might have been influenced by these factors. Furthermore, these authors also observed that the use of anti-*Leishmania* monoclonal antibodies led to a decrease in the sensitivity of IHC to 71%, and the polymer detection kit performed similarly to the streptavidin-biotin-peroxidase detection kit.

The use of IHC for the detection of amastigotes in NWCL lesions in dogs caused by *L.* (*V.*) *braziliensis* increased the diagnostic sensitivity to 70.0%, compared with 37.5% by HP [29]. Menezes et al. [13] assessed the sensitivity of IHC in samples of intact skin from dogs infected by *L. infantum*, and obtained a sensitivity of 69.5% using a protocol similar to the one described herein. Although the sensitivity found in the present study was slightly lower (66%), it is noteworthy that the parasite load described in skin samples is usually high in canine visceral leishmaniasis when compared with human NWCL. Still, the validation and standardization of these protocols in reference laboratories are essential to obtain more reliable and reproducible diagnosis methods, so that differences in sensitivity will not be attributed to inter-laboratory variations.

Another aspect worth mentioning is that the sensitivity assessment of CISH for the diagnosis of NWCL caused by *L.* (*V.*) *braziliensis* is being reported for the first time herein. This sensitivity was within the range expected for parasitological diagnosis of NWCL, being slightly superior to that of HP. However, as with IHC and HP, CISH is less sensitive than PCR, which shows a sensitivity of 87 to 98% [5,30,31,32,33,34]. This lower sensitivity of in situ hybridization compared with PCR is due to the fact that there is no amplification of parasite DNA in this technique as in PCR, but rather a detection of nucleic acids in the sample [11,14]. Nonetheless, the main advantages of in situ hybridization compared with PCR are the abilities to diagnose active infections and allow for the correlation between the presence of the parasite and the lesion [35]. In addition, PCR may have inaccuracies in sensitivity, especially in FFPE tissues [8,9], and a combination with a parasitological or histological technique is recommended to increase accuracy in the diagnosis of human NWCL [3]. Another drawback of PCR is that it is more expensive than parasitological and histological techniques due to the necessary equipment, reagents, infrastructure, and the sample processing complexity [2,10]. However, further cost-effectiveness studies of the techniques used in the present study and of PCR for the diagnosis of human NWCL are recommended to help choose the best approaches for the diagnosis of this disease.

In contrast with CISH, FISH has already been tested for the detection of amastigote forms of *Leishmania* spp. in FFPE skin samples from human patients using generic probes for *Leishmania* spp. different from the one used in the present study [12,36]. Frickmann et al. [12] detected *L. major*/*tropica* amastigotes in four out of eleven (36%) FFPE skin samples from human patients with a clinical diagnosis of old world cutaneous leishmaniasis (OWCL). On the other hand, Kaluarachchi et al. [36], using the same generic probes developed by Frickmann et al. [12], identified a higher sensitivity (80.9%), compared with HP (50.4%) for the detection of *Leishmania* amastigotes in FFPE skins from Sri Lankan patients with OWCL caused by *L. donovani*. These different sensitivity values found by Frickmann et al. [12] and Kaluarachchi et al. [36], compared with those of the present study, were probably related to the use of FISH with different probes and the diagnosis of positive skin samples for old world *Leishmania* species.

Menezes et al. [13] found a sensitivity of 70% in the diagnosis of *L. infantum* in the intact skins of dogs using the same probe as the one employed in this study and the CISH technique in an automated protocol. This sensitivity was lower than that found with the use of a specific probe for *L. infantum* (74.5%) and higher than the sensitivity of IHC (69.5%) and histopathology (57.6%). In the automated protocol described by Menezes et al. [13], the *Leishmania* species and host investigated may have acted upon better sensitivity results found by CISH compared with those in the present study. Another factor that may have influenced the lower CISH sensitivity results obtained in the present study was the longer storage time of the blocks—over 6 years—compared with the storage time of the blocks in the study of Menezes et al. [13], which was up to 4 years. This longer storage time in the present study may have degraded the target DNA, impairing the sensitivity of the CISH technique [37,38]. The results of higher CISH sensitivity and higher agreement between CISH and IHC in the younger blocks (12 to 13 years of storage) compared with the older blocks (14 to 17 years of storage) in the present study reinforce this hypothesis. Moreover, the lower sensitivity found by CISH compared with that by IHC probably resulted from greater tissue damage caused by heat treatment and protein digestion, different from that used in IHC. In the IHC protocol used in this study, heat treatment was used for antigen retrieval at lower temperature (65°C) and protein digestion was not performed, which promoted less tissue damage, allowing for the better visualization of the stained amastigote forms. A limitation of the present study is that the parasite load in the cutaneous lesions was not quantified, and this variable may have influenced not only the sensitivity values obtained by CISH, HP, and IHC, but also the agreement among these techniques and observers.

Even though HP showed the lowest sensitivity, it was considered useful for the detection of amastigote forms of *L.* (*V.*) *braziliensis*. Other studies found similar sensitivity for the diagnosis of human NWCL [10,25]. In addition, this method has a lower cost and it is faster, requiring a less complex laboratory structure than IHC and CISH and allowing for the visualization of microscopic lesions. Despite being specific, the reading of organisms with similar histomorphologies such as *Histoplasma* spp. and *Sporothrix* spp. may be unclear. According to the literature and the results showed herein, to increase sensitivity in the histological diagnosis and confirmation of cases, it is key to perform IHC or CISH [10,13].

In this study, IHC presented sensitivity results greater than those found by CISH for the diagnosis of NWCL in FFPE samples. Nevertheless, IHC showed cross-reaction with the three species of fungi included in the study, as they are histomorphologically similar to *Leishmania* spp. This cross-reactivity with fungi in IHC for the diagnosis of NWCL using polyclonal rabbit anti-*Leishmania* sp. serum was reported by other authors for *Paracoccidioides brasiliensis* [10,22,25], *Sporothrix* sp. [25], and *Histoplasma capsulatum* [10]. Menezes et al. [13], using a protocol similar to that of IHC for diagnosing *L. infantum* in the intact skins of dogs, found a cross-reaction for *H. capsulatum* and *Trypanosoma cruzi*. Due to the fact that a polyclonal rabbit serum obtained in-house was the primary antibody used, this may have impaired the technique employed.

Unlike IHC, there was no cross-hybridization in CISH with fungi histomorphologically similar to the amastigote forms of *Leishmania* sp. [13]. Other authors obtained similar results and considered CISH to be more specific than IHC, as it detects parasite-specific nucleic acid sequences and not antigens [14,37].

According to the three observers and based on the values of the Cohen kappa index, HP revealed the best agreement (almost perfect and substantial), followed by CISH (almost perfect and substantial, but with lower values compared with HP). On the other hand, IHC showed the lowest agreement, especially when comparing observer one with observers two and three (moderate agreement). This finding, moderate agreement in IHC, may be due to the brown-stained amastigote forms conferred by DAB chromogen used in this technique, which can be mistaken by the brown staining of intracellular pigments, such as hemosiderin and melanin [39]. The use of other chromogens that give colorimetric reactions different from the one employed, such as NBT and BCIP (intense blue staining) or Fast Red chromogen (red staining), may prevent this when reading the slides. The CISH protocol in this study used NBT/BCIP chromogens, which may have provided better inter-observer agreement than IHC. Another hypothesis would be observer expertise, as reported for the detection of amastigote forms of *Leishmania* sp. in cytopathology and IHC [40]. Accordingly, observer’s training would be a measure to minimize discrepancies among the reading results, especially in IHC.

## 5. Conclusions

IHC shows greater sensitivity to the histological diagnosis of *L.* (*V.*) *braziliensis* in FFPE samples of cutaneous lesions from patients with NWCL when compared with CISH and the traditional HP method, thus meaning that its use improves diagnosis accuracy. Nevertheless, the reading of the IHC slides was more subject to inter-observer variation, suggesting that the observer’s training and the change in the chromogen used in the technique are recommended. The results point out CISH as a more specific method than IHC for the diagnosis of human NWCL in FFPE samples of cutaneous lesions, and therefore it can be used in combination with IHC to improve the diagnosis of this disease

## Figures and Tables

**Figure 1 tropicalmed-07-00344-f001:**
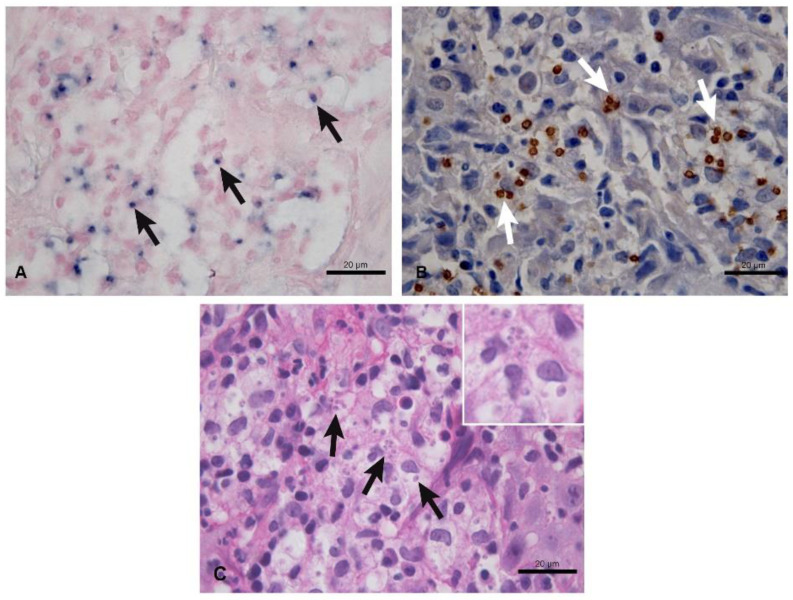
Positive staining for amastigotes in samples fixed in 10% formalin and paraffin-embedded of cutaneous lesions from patients with new world cutaneous leishmaniasis caused by *L.* (*V.*) *braziliensis*. (**A**) Dark-blue-stained amastigote forms (arrows) inside macrophages. Colorimetric in situ hybridization, 100× objective. (**B**) Brown-stained amastigote forms (arrows) inside macrophages. Immunohistochemistry, 100× objective. (**C**) Amastigote forms inside the parasitophorous vacuoles of macrophages (arrows and inset) in the dermis amidst an inflammatory reaction consisting mainly of macrophages, with fewer lymphocytes and plasma cells and rare neutrophils. Histopathology, 100× objective.

**Figure 2 tropicalmed-07-00344-f002:**
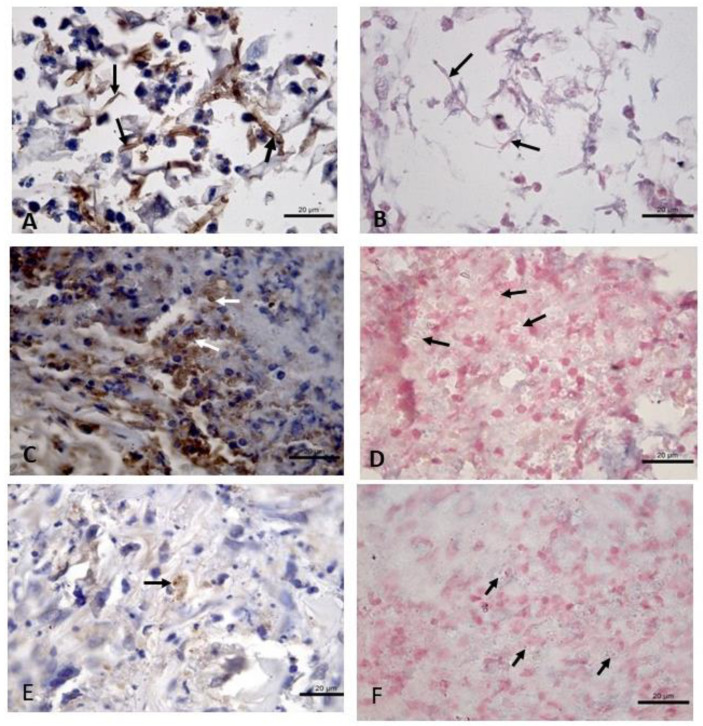
Cross-reaction evaluation by immunohistochemistry (IHC) using polyclonal rabbit anti-*Leishmania* sp. serum and cross-hybridization by colorimetric in situ hybridization (CISH) using a generic probe for *Leishmania* spp. in samples fixed in 10% formalin and paraffin-embedded of cutaneous lesions from fungal-infected patients. (**A**) Infected skin by *Candida* sp. Brown stained hyphae are observed (arrows). IHC, 100× objective. (**B**) Infected skin by *Candida* sp. There was no staining of hyphae using the generic probe for *Leishmania* spp. (arrows). CISH, 100× objective. (**C**) Infected skin by *Sporothrix* sp. Brown-stained yeasts are observed (arrows). IHC, 100× objective. (**D**) Infected skin by *Sporothrix* sp. There was no staining of yeasts using the generic probe for *Leishmania* spp. (arrows). CISH, 100× objective. (**E**) Infected skin by *Histoplasma* sp. showing brown-stained yeasts. (arrow). IHC, 100× objective. (**F**) Infected skin by *Histoplasma* sp. There was no cross-staining using the generic probe for *Leishmania* spp. (arrows). CISH, 100× objective.

**Table 1 tropicalmed-07-00344-t001:** Sensitivity for the detection of amastigote forms found by IHC, CISH, and HP and the number of positive and negative samples for each of these techniques were assessed for 50 samples fixed in 10% formalin and paraffin-embedded cutaneous lesions from patients with new world cutaneous leishmaniasis caused by *L.* (*V.*) *braziliensis*.

Technique	Sensitivity (CI 95%)	Positive Samples	Negative Samples
IHC	66.0% (51.2–78.8%)	33	17
CISH	54.0% (39.3–68.2%)	27	23
HP	50.0% (35.5–64.5%)	25	25

CISH: colorimetric in situ hybridization; IHC: immunohistochemistry; HP: histopathology; CI: confidence interval.

**Table 2 tropicalmed-07-00344-t002:** Sensitivity for the detection of amastigote forms found by IHC, CISH, and HP according to the time of storage of 50 paraffin blocks from patients with new world cutaneous leishmaniasis caused by *L.* (*V.*) *braziliensis*.

Time of Storage (Years)	N	Sensitivity (CI 95%)		
		IHC	CISH	HP
14 to 17	20	70.0% (45.7–88.2%)	40.0% (19.1–64.0%)	50.0% (27.2–72.8%)
12 to 13	30	63.3 % (43.9–80.0%)	66.7% (47.2–82.7%)	50.0% (31.3–68.7%)

N: number of samples; IHC: immunohistochemistry; CISH: colorimetric in situ hybridization; HP: histopathology; CI: confidence interval.

**Table 3 tropicalmed-07-00344-t003:** Comparison of the number of positive and negative cases for amastigotes obtained using CISH, IHC, and HP in samples fixed in 10% formalin and paraffin-embedded of cutaneous lesions from patients with new world cutaneous leishmaniasis caused by *L.* (*V.*) *braziliensis*.

Technique		IHC		HP	
		Positive n (%)	Negative n (%)	Positive n (%)	Negative n (%)
CISH	Positive	24 (48%)	3 (6%)	19 (38%)	8 (16%)
	Negative	9 (18%)	14 (28%)	6 (12%)	17 (34%)
HP	Positive	23 (46%)	10 (20%)	25 (50%)	0
	Negative	2 (4%)	15 (30%)	0	25 (50%)

CISH: colorimetric in situ hybridization; IHC: immunohistochemistry; HP: histopathology.

**Table 4 tropicalmed-07-00344-t004:** Agreement between diagnosis techniques for the detection of amastigotes in samples fixed in 10% formalin and paraffin-embedded of cutaneous lesions from patients with new world cutaneous leishmaniasis caused by *L.* (*V.*) *braziliensis*.

Comparison of Techniques *	Cohen Kappa Index	Agreement ^a^	Total Agreement
CISH vs. IHC	0.51 (CI 0.27–0.74)	Moderate	76%
CISH vs. HP	0.44 (CI 0.19–0.69)	Moderate	72%
IHC vs. HP	0.52 (CI 0.3–0.74)	Moderate	76%

CISH: colorimetric in situ hybridization; IHC: immunohistochemistry; HP: histopathology; CI: 95% confidence interval; vs: versus; * considering the data by observer 1. ^a^ Level of agreement according to Landis and Koch [21].

**Table 5 tropicalmed-07-00344-t005:** Agreement between diagnosis techniques for the detection of amastigotes in samples fixed in 10% formalin and paraffin-embedded of cutaneous lesions according to the time of storage of paraffin blocks from patients with new world cutaneous leishmaniasis caused by *L.* (*V.*) *braziliensis*.

Techniques *	Time of Storage of Paraffin Blocks					
	14 to 17 Years			12 to 13 Years		
	Cohen Kappa	Agreement ^a^	Total Agreement	Cohen Kappa	Agreement ^a^	Total Agreement
CISH vs. IHC	0.37 (CI 0.08–0.67)	fair	65.0%	0.63 (CI 0.34–0.92)	substantial	83.3%
CISH vs. HP	0.50 (CI 0.14–0.86)	moderate	75.0%	0.40 (CI 0.09–0.71)	moderate	70.0%
IHC vs. HP	0.60 (CI 0.28–0.92)	substantial	80.0%	0.47 (CI 0.16–0.77)	moderate	73.3%

CISH: colorimetric in situ hybridization; IHC: immunohistochemistry; HP: histopathology; CI: 95% confidence interval; vs: versus; Cohen kappa index: 0.21–0.40 (fair); 0.40–0.59 (moderate); Cohen kappa index: 0.60–0.79 (substantial); Cohen kappa index: 0.80–1.00 (almost perfect); * considering the data by observer 1. ^a^ Level of agreement according to Landis and Koch [21].

**Table 6 tropicalmed-07-00344-t006:** Inter-observer agreement for the IHC, CISH, and HP techniques for the detection of amastigotes in skin samples fixed in 10% formalin and paraffin-embedded from patients with new world cutaneous leishmaniasis caused by *L.* (*V.*) *braziliensis*.

Technique	Observer 1 vs. 2		Observer 1 vs. 3		Observer 2 vs. 3	
	Cohen Kappa	Agreement ^a^	Cohen Kappa	Agreement ^a^	Cohen Kappa	Agreement ^a^
CISH	0.80 (IC 0.64–0.97)	almost perfect	0.68 (CI 0.49–0.88)	substantial	0.64 (CI 0.42–0.85)	substantial
IHC	0.46 (IC 0.21–0.72)	moderate	0.58 (CI 0.35–0.81)	moderate	0.75 (CI 0.56–0.93)	substantial
HP	0.76 (IC 0.58–0.94)	substantial	0.72 (CI 0.53–0.91)	substantial	0.88 (CI 0.75–1.00)	almost perfect

CISH: colorimetric in situ hybridization; IHC: immunohistochemistry; HP: histopathology; CI: 95% confidence interval; vs: versus; Cohen kappa index: 0.40–0.59 (moderate); Cohen kappa index: 0.60–0.79 (substantial); Cohen kappa index: 0.80–1.00 (almost perfect). ^a^ Level of agreement according to Landis and Koch [21].

## Data Availability

The data presented in this study are available upon request.

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
