# Peer review of "Comparison between Colorimetric In Situ Hybridization, Histopathology, and Immunohistochemistry for the Diagnosis of New World Cutaneous Leishmaniasis in Human Skin Samples"

_tropicalmed, 2022, doi:10.3390/tropicalmed7110344_

Round 1

Reviewer 1 Report

see attached file

Author Response

Responses to reviewer 1

The English language of the manuscript was revised, as recommended.

Abstract. Line 39. Suggest to change it to complementary assay, this technique is less sensitive then your IHC. For diagnostic is preferable a more sensitive test.

Response: The text “good alternative” was replaced with “complementary assay”, as recommended.

Introduction. Line 58.

Response:

We wrote "for fresh specimens" after "sensitivity limitations" as recommended.

Materials and methods

Item 2.6. Line 170. Why did not use the consensus result among the two more experienced pathologist?

Response:

We did not used the consensus among the observers, as the observer 1 was not only the most experienced observer, but also the only physician pathologist among the observers.  The inclusion of the observers 2 and 3 in the study was to verify the ease of reading the slides obtained by the different techniques evaluated.

Therefore, we maintained the text.

We included the word physician before pathologist to make the text clearer.

Item 2.6. Line 181. Only immunostaining was taken in consideration for cross reactivity? How about amastigote size and shape, please clarify.

Response:

We replaced the text “Tissue samples containing fungal forms to investigate cross-reaction or cross-hybridization were considered positiveif brown or dark blue staining was observed by the chromogen used in IHC and CISH, respectively” with “ For fungal-containing tissue samples to be considered positive for cross-reactivity or cross-hybridization, both characteristics must be present: morphologically compatible with yeasts or hyphae and brown or dark blue stained by the chromogen used in IHC and CISH, respectively.”

Item 2.7. Line 185. My suggestion again is to include the results from two independent observers for sensitivity calculation

Response:

Thanks for the suggestion. We maintained the text as justified above for the item 2.6

Item 3.  Results

 Figure 1. Please include the magnification used for these pictures

Response:

We have included the magnification of the figures (100X objective).

Line 216.

Response:

The verb “illustrates” was replaced with the verb “presents”.

Line 237. Again, it is taking in consideration only the staining not morphological parameters for this statement, correct?

Response:

As we explained above, we did not taken in consideration not only the staining but also de morphology of the fungi that were tested.

In order to make the text clearer, we replaced the text: “CISH showed no cross-hybridization with Sporothrix sp., Candida sp., and Histoplasma sp., whereas IHC presented cross-reactions with all these three fungal species herein evaluated (Figure 2).” with “CISH showed no cross-hybridization with structures morphologically compatible with yeasts or hyphae of Sporothrix sp., Candida sp., and Histoplasma sp., whereas IHC showeded cross-reactions with structures morphologically compatible with yeasts or hyphae of the three fungal species herein evaluated (Figure 2).”

Line 240. Figure 2.

 Please include magnification.

Response:

We have included the magnification of the figures (100X objective).

Item 4. Discussion

Line 287. We wrote the word "results" after "sensitivity".

Line 275. As well different protocol used?

Response:

We have included the following text after the word lesions: "...as well as the different protocol used."

Line 321. Less agreement was found on older blocks? This data was not presented. How was the agreement results based on the block age?

Response:

 There were a higher sensitivity results of CISH and a higher agreement between CISH and IHC in the younger blocks (12 to 13 years of storage) compared to the older blocks (14 to 17 years of storage). Therefore, we have included in the Results section the Tables 2 and 5. Table 2 shows the frequency of positivity for the IHC, CISH and HP techniques in the samples in relation to the storage time of the blocks. Table 5 shows the agreement between diagnosis techniques for the detection of amastigotes in samples fixed in 10% formalin and paraffin-embedded of cutaneous lesions according to the time of storage of paraffin blocks.

In addition, we wrote the following text in the discussion:

 “Another factor that may have influenced the lower CISH sensitivity results obtained in the present study was the longer storage time of the blocks -over 6 years- compared to the storage time of the blocks in the study of Menezes et al. [13], which was up to 4 years. This longer storage time in the present study may have degraded the target DNA probe, impairing the sensitivity of the CISH technique [37, 38]. The results of higher CISH sensitivity and higher agreement between CISH and IHC in the younger blocks (12 to 13 years of storage) compared to the older blocks (14 to 17 years of storage) in the present study reinforce this hypothesis.”

In the materials and methods section, item 2.1, we included the following text:

“One of the skin samples was from 2000, another from 2001, eight were from 2002, ten from 2003, 18 from 2004, and 12 from 2005.”

Line 324. How about parasite load in the tissue? Please include this information on the results. High discordances were observed when low parasitism was observed also? Please include this discussion here.

Response:

One limitation of the present study is that parasite load was not quantified in the cutaneous lesions and this variable may have influenced not only on the sensitivity values obtained buy CISH, HP and IHC, but also on the agreement among these techniques and inter-observers.

 Therefore, we have included the following text:

“A limitation of the present study is that the parasite load in the cutaneous lesions was not quantified, and this variable may have influenced not only the sensitivity values obtained by CISH, HP and IHC, but also the agreement among these techniques and observers. “

Discussion. Lines 346-347. IHC more sensitivity but less specificity, CISH more specificity but less sensitivity then IHC technique presented here. Complementary diagnostic methodologies.

Response:

We agree with the suggestion and thus we changed the conclusions as shown below (item 5).

5.Conclusions.

 CISH is a more specific method for the identification of amastigotes in the cutaneous lesion and can be used in combination with IHC technique to improve the identification of NWCL

Response:

The text: “The results pointed out CISH as an accurate method for the diagnosis of human NWCL in FFPE samples of cutaneous lesions and therefore enabling a new tool to help diagnosis.” was replaced with “The results pointed out CISH as a more specific method than IHC for the diagnosis of human NWCL in FFPE samples of cutaneous lesions and therefore can be used in combination with IHC to improve the diagnosis of this disease.”

Reviewer 2 Report

The study compares three diagnostic methods for NWCL in human cutaneous lesions. The paper is well written with no errors in language

As per manuscript title the study results are acceptable and can be published. Considering the disease burden in the area the available diagnostic test should be quick, reliable, sensitive and cost effective. Therefore please address the following suggestions:

The primary objective of a diagnostic test is to detect the disease of a  patient with the cheapest and most sensitive diagnostic test available. All three tests that are compared here, the sensitivity is very much less than that of PCR (gold standard) where the sensitivity is around 98%. It is well known that the sensitivity of HP is low and is always followed up by PCR for confirmation.

In addition what is the cost effectiveness of these tests as compared to PCR? Is it worth recommending a test that is less sensitive and more expensive than what is available currently  

Author Response

Please, see attachment.

Reviewer 2

 Comments and Suggestions for Authors

The study compares three diagnostic methods for NWCL in human cutaneous lesions. The paper is well written with no errors in language

As per manuscript title the study results are acceptable and can be published. Considering the disease burden in the area the available diagnostic test should be quick, reliable, sensitive and cost effective. Therefore please address the following suggestions:

The primary objective of a diagnostic test is to detect the disease of a patient with the cheapest and most sensitive diagnostic test available. All three tests that are compared here, the sensitivity is very much less than that of PCR (gold standard) where the sensitivity is around 98%. It is well known that the sensitivity of HP is low and is always followed up by PCR for confirmation.

In addition what is the cost effectiveness of these tests as compared to PCR? Is it worth recommending a test that is less sensitive and more expensive than what is available currently?

Response:

There is not a single reference test for the diagnosis of NWCL. The PCR is more sensitive, but have inaccuracies in sensitivity, notably in cases of strong cellular immune response or when performed in 10% formalin-fixed paraffin-embedded tissues. In addition, PCR does not diagnose active infection and allow for the correlation between the presence of the parasite and the lesion. Another drawback of PCR is that it is considered more expensive than parasitological and histological techniques due to equipment, reagents and infra-structure that are necessary and complexity of sample processing.

The present study did not evaluate the cost-effectiveness of the tests evaluated, which would require a specific methodology. The cost-effectiveness of the tests for the diagnosis of NWCL is little know. Therefore, further cost-effectiveness studies of the techniques used in the present study and PCR for the diagnosis of human NWCL are recommended to help in the choose of the best approaches for the diagnosis of this disease. Therefore, a combination of PCR with a parasitological or histological technique is recommended for increasing the accuracy for the diagnosis of human NWCL.

In order to make the text clearer about the drawbacks of PCR and its cost compared to other techniques for the diagnosis of human NWCL in the literature, we included the following text in the discussion:

Discussion.

“In addition, PCR may have inaccuracies in sensitivity, especially in FFPE tissues [8, 9] and a combination with a parasitological or histological technique is recommended to increase accuracy in the diagnosis of human NWCL [3]. Another drawback of PCR is that it is more expensive than parasitological and histological techniques due to the necessary equipment, reagents, infrastructure and sample processing complexity [2,10]. However, further cost-effectiveness studies of the techniques used in the present study and of PCR for the diagnosis of human NWCL are recommended to help choose the best approaches for the diagnosis of this disease.”
